# A Spike-like Self-Assembly of Polyaspartamide Integrated with Functionalized Nanoparticles

**DOI:** 10.3390/polym16020234

**Published:** 2024-01-15

**Authors:** Sa Ra Han, Yujin Ahn, Sungwoo Cho, Hyewon Jeong, Yoonsook Ji, Woonggyu Jung, Jae Hyun Jeong

**Affiliations:** 1Department of Chemical Engineering, Soongsil University, Seoul 06978, Republic of Korea; srhan@ssu.ac.kr (S.R.H.); chosw1120@gmail.com (S.C.); hw1205d@soongsil.ac.kr (H.J.); ysjji53@soongsil.ac.kr (Y.J.); 2Department of Biomedical Engineering, Ulsan National Institute of Science and Technology (UNIST), Ulsan 44919, Republic of Korea; ahn.unist@gmail.com

**Keywords:** spike-like self-assembly, polyaspartamide, UV blocking, functional nanoparticles

## Abstract

The integration of nanoparticles (NPs) into molecular self-assemblies has been extensively studied with the aim of building well-defined, ordered structures which exhibit advanced properties and performances. This study demonstrates a novel strategy for the preparation of a spike-like self-assembly designed to enhance UV blocking. Poly(2-hydroxyethyl aspartamide) (PHEA) substituted with octadecyl chains and menthyl anthranilate (C_18_-M-PHEA) was successfully synthesized by varying the number of grafted groups to control their morphology and UV absorption. The in situ incorporation of polymerized rod-like TiO_2_ within the C_18_-M-PHEA self-aggregates generated spike-like self-assemblies (TiO_2_@C_18_-M-PHEA) with a chestnut burr structure in aqueous solution. The results showed that the spike-like self-assemblies integrated with TiO_2_ NPs exhibited a nine-fold increase in UV protection by simultaneous UV absorption and scattering compared with the pure TiO_2_ NPs formed via a bulk mixing process. This work provides a novel method for UV protection using self-assembling poly(amino acid)s derivatives integrated with functional nanoparticles to tune their morphology and organization.

## 1. Introduction

Molecular self-assemblies integrated with functional nanoparticles (NPs) are being extensively researched to enhance the properties and performances of materials used in various applications such as cosmetics, medicine, and agriculture [1,2,3]. In particular, 3D organic–inorganic hybrid systems formed by the self-assembly of amphiphilic copolymers with functional metallic NPs offer advantages such as size control, enhanced stability, and additional functionalities [4,5,6]. Previous studies have demonstrated the thermodynamically favored micelle-to-vesicle transition through the in situ polymerization of platinum NPs within polyaspartamide self-aggregates, showcasing improved functionality [3]. The successful use of molecular self-assemblies in biological and industrial applications greatly relies on the ability to direct polymeric self-assembly integrated with NPs while imparting the desired functionality. In general, however, the incorporation of nanoparticles into polymeric self-aggregates often results in the irregular arrangement of polymers and further phase separation, but limited efforts have been made to resolve this challenge, specifically in cosmetics and organic and inorganic emulsion systems.

Therefore, our study introduces an innovative strategy to engineer spike-like molecular self-assembly specifically tailored to enhanced UV-blocking capabilities (Figure 1b). Given that UV rays are a primary cause of skin aging, the demand for effective sunscreens has surged [7,8]. Conventional methods of preparing inorganic scattering sunscreens often incur additional costs, as preventing chalky white layers resulting from aggregation is essential for maintaining optimal performance [9,10].

Our hypothesis centers on leveraging the spike-like self-assembly of poly(amino acid) derivatives integrated with TiO_2_ NPs to achieve superior UV blocking while mitigating undesired aggregations [11,12]. We put this hypothesis to the test by synthesizing poly(2-hydroxyethyl aspartamide) (PHEA) substituted with octadecyl chains and menthyl anthranilate, strategically varying the number of grafted groups to exert control over their morphology and UV absorption. The in situ incorporation of polymerized rod-like TiO_2_ within the PHEA self-aggregates led to the formation of spike-like self-assemblies, characterized by a distinctive chestnut burr structure in aqueous solution. As depicted in Figure 1, this intricate morphology, akin to the natural arrangement of chestnut burrs, highlights the exceptional organization achieved within the self-assemblies, contributing to their enhanced functionality for UV-blocking applications. The spike-like configuration, coupled with the chestnut burr structure, symbolizes a synergistic and precisely engineered architecture that holds great promise for improving the performance and versatility of these self-assemblies in various practical applications. To evaluate the efficacy of this new strategy, we examined the resulting spike-like self-assemblies integrated with TiO_2_ NPs by assessing UV blocking through simultaneous UV absorption and scattering. This comparative analysis involved the pure TiO_2_ NPs formed via a bulk mixing process. Our findings not only contribute to the advancement of molecular self-assembly with functional NPs but also hold promise for refining their functionality by precisely tuning their morphology and organization [13,14,15,16].

## 2. Experiments

### 2.1. Synthesis of Poly(2-hydroxyethyl aspartamide) Substituted with Octadecyl Chains and Menthyl Anthranilate (C_18_-M-PHEA)

Figure 1a illustrates the overall synthesis scheme of M-PHEA and C_18_-M-PHEA. First, polysuccinimide (PSI) was synthesized by acid-catalyzed thermal polycondensation of L-aspartic acid [17]. Briefly, L-aspartic acid and phosphoric acid were suspended in cosolvents (200 g; sulfolane to mesitylene ratio of 3:7) and mechanically stirred at 170 °C under a N_2_ atmosphere. The water generated during the reaction was eliminated using a Dean-Stark trap apparatus. The reaction mixture was precipitated in excess methanol after 10 h and washed with deionized water until the mixture became neutral. The synthesized PSI was washed with methanol and dried at 80 °C in a vacuum oven. The molecular structure of PSI was confirmed by ^1^H NMR analysis. The signals at 2.7, 3.2, and 5.3 ppm were attributed to the methylene and methane protons of the PSI unit, respectively (Appendix A). Then, PSI solution was prepared by dissolving the as-prepared PSI (970 mg) in DMSO (10 mL, Sigma, St. Louis, MO, USA). Octadecylamine (C_18_, 135 mg) was dissolved in DMF (1 mL, Sigma, St. Louis, MO, USA) and added to the solution. The aminolysis reaction mixture was then stirred at 80 °C in an oil bath. After 48 h, specified amounts of menthyl anthranilate (MA, Sigma, St. Louis, MO, USA), used as a UVA-absorbing ingredient, were dissolved in DMF (1 mL) and then added to the mixture. The grafting reaction was maintained at 80 °C for 48 h in an oil bath and then cooled to room temperature. Finally, ethanolamine was added to the solution to prepare the C_18_-M-PHEA via a ring-opening mechanism. After stirring at room temperature for 12 h, the solution was transferred to a rotary evaporator to remove the solvent. The mixture was precipitated in excess methanol thrice and dried under vacuum at 60 °C. The resulting chemical structure of C_18_-M-PHEA was confirmed by ^1^H NMR spectroscopy. M-PHEA was synthesized using the same method, omitting C_18_ synthesis, and the molecular structure was confirmed by ^1^H NMR.

The cellular metabolic activities of the synthesized M-PHEA and C_18_-M-PHEA solutions were analyzed after 24, 48, and 72 h, following the treatment, to verify toxicity. Human dermal fibroblasts (HDFs) were used for toxicity tests to align with the primary purpose of the material in the transdermal drug delivery development. The HDF cells (Gibco, Miami, FL, USA) were cultured in a 75 cm^2^ flask using Dulbecco’s Modified Eagle’s Medium (DMEM, Biowest, Nuaillé, France) with 10% fetal bovine serum (FBS, Biowest, Nuaillé, France) and 1% penicillin-streptomycin (PS, Biowest, Nuaillé, France) in a 5% CO_2_ incubator at 37 °C. Then, the HDFs were seeded in 96-well plates (10,000 cells each) and the synthesized M-PHEA and C_18_-M-PHEA polymer solution in phosphate-buffered saline (PBS, Biowest, Nuaillé, France) was dropped in each well at pre-determined concentrations. After incubation at 37 °C for a specified time, the mixture was removed. Next, 100 μL of fresh medium and 10 μL of diluted MTT solution (5.0 mg/mL) were added to each well in low-light conditions. The lysis buffer solution (100 μL) was added after 4 h and incubated for 2 h. The cell lysis solutions were used to quantify the optical density (OD) at 570 nm [18].

### 2.2. Self-Assembly of M-PHEA and C_18_-M-PHEA in an Aqueous Solution

Stable solutions of self-assembled M-PHEA and C_18_-M-PHEA were prepared via precipitation and dialysis. Purified copolymers (10 mg) were dissolved in DMSO (1.0 mL), which is a good solvent for both PHEA and C_18_, deionized water (10 mL) was added to induce the assembly of the graft copolymers, and the mixture was further sonicated for five minutes using a tip-type sonifier at room temperature. The resulting suspensions were stirred and then dialyzed for 3 days to remove the remaining DMSO using a dialysis membrane (MWCO = 8000~12,000 g/mol). The size of the self-assembled particles was measured using a dynamic light scattering (DLS, ZS90, Malvern Instruments Ltd., Worcestershire, UK) equipped with a He-Ne Laser of wavelength 520 nm at a scattering angle of 90° (by the Stokes–Einstein relationship). The critical aggregation concentration of polymers was measured using a pyrene probe. The aqueous polymer suspension mixed with pyrene (6.0 × 10^−7^ mol/L, Sigma) was excited at a wavelength of 330 nm, and the resulting emission spectrum was obtained using a fluorometer (QM40, PTI, Birmingham, NJ, USA). The bandwidth was adjusted to 2.0 nm for both excitation and emission. The morphology of self-assembled polymeric nanoparticles was also analyzed using transmission electron microscopy. A drop of nanoparticle suspension was placed on a copper grid coated with carbon film. The images were captured at 120 kV with TEM (JEM-2010 HC, JEOL, Peabody, MA, USA).

### 2.3. Self-Assembly of C_18_-M-PHEA Integrated with TiO_2_ NPs

First, TiO_2_ NPs were synthesized using precursors, titanium tetra-isopropoxide (TTIP, Sigma, Sigma, St. Louis, MO, USA). TTIP was dissolved in propanol with aluminum hydroxide at room temperature, and stearic acid was dissolved in propanol at 50 °C. The TTIP solution was then added to the stearic acid solution. This mixture was stirred for 20 min until it was sufficiently mixed and then centrifuged at 1000 rpm for 5 min. The precipitate was separated by decantation of the supernatant and dried at 170 °C for 1 h. Finally, the TiO_2_ powder was obtained after calcination at 400 °C for 4 h. The structure of the synthesized TiO_2_ was characterized using X-ray diffraction (XRD). Then, in situ incorporation of synthesized TiO_2_ NPs within the C_18_-M-PHEA self-aggregates was performed using the emulsification method. TiO_2_ NPs (20 mg) dissolved in chloroform (300 μL) were added to an aqueous solution of self-assembled M-PHEA and C_18_-M-PHEA. The solution was then stirred at 3000 rpm for 5 min and immediately sonicated for 10 min at low temperature. The chloroform in the polymer emulsion solution was evaporated using a rotary evaporator. The morphology of the self-assembly of C_18_-M-PHEA integrated with TiO_2_ NPs was also analyzed using transmission electron microscopy (JEM-2010 HC, JEOL, Peabody, MA, USA) at 120 kV. A drop of self-assembly solution containing 0.1% phosphotungstic acid was placed on a copper grid coated with carbon film. The grid was held horizontally for 30 s to allow the aggregates to settle and then vertically to allow excess fluid to drain. Subsequently, the grid was carefully dried, and the prepared sample was ready for TEM analysis. UV-blocking evaluation was conducted by measuring the UV absorbance using a UV/Vis spectrophotometer (JASCO, V-650, Easton, MD, USA).

## 3. Results and Discussion

### 3.1. Synthesis and Self-Assembly of C_18_-M-PHEA

First, we synthesized poly(2-hydroxyethyl aspartamide) (PHEA) substituted with octadecyl chains (C_18_) and menthyl anthranilate (MA), termed C_18_-M-PHEA, by modifying polysuccinimide (PSI) in a top-down manner. The PSI precursor was prepared via the acid-catalyzed polycondensation of L-aspartic acid. Then, designated amounts of C_18_, MA, and 2-ethanol amine were sequentially added to the PSI solution to prepare C_18_-M-PHEA through nucleophilic substitution to PSI (Figure 1a(2)). The presence of C_18_ and MA in the C_18_-M-PHEA was confirmed by ^1^H NMR spectroscopy (Table 1 and Figure 2a). The degree of substitution (DS) of C_18_ and MA to PHEA (DS_C18_ and DS_M_), defined as the mole percentage of succinimide units substituted with C_18_ and MA, respectively, was varied by altering the molar ratio between the grafts and the succinimide units of PSI. Additionally, PHEA substituted with MA, termed M-PHEA, was synthesized and confirmed to be compared with C_18_-M-PHEA in self-assembly (Figure 1a(1) and Figure 2a). As illustrated in Table 1, the final synthesized M-PHEA series includes M10-PHEA, M20-PHEA, and M30-PHEA, where the feed DS_M_ of MA is 10, 20, and 30 mol%, respectively (Appendix A). Furthermore, the C_18_-M-PHEA series maintains a consistent feed DS_C18_ of 5 mol% for C_18_, while the DS of MA is varied to create C_18_-M10-PHEA and C_18_-M20-PHEA. Through this approach, we aimed to investigate the influence of the amount of grafted MA on the self-assembly behavior of M-PHEA series in aqueous solutions. Additionally, the preparation of C_18_-M-PHEA polymers with the introduction of TiO_2_ nanoparticles aimed to impart sufficient hydrophobicity during the self-assembly process.

Then, self-assemblies of M-PHEA and C_18_-M-PHEA were prepared by precipitation and dialysis in aqueous solutions. M-PHEA and C_18_-M-PHEA copolymers were self-assembled in aqueous solution due to the intra- and/or intermolecular hydrophobic interactions of the grafted hydrophobic segments [19]. The size distribution of the aggregates of M-PHEA and C_18_-M-PHEA in aqueous solution (0.5 mg/mL) was measured by DLS (dynamic light scattering) (Figure 3). DLS is a non-invasive technique that measures the hydrodynamic diameter of particles in a solution by analyzing the intensity fluctuations in scattered light by the Stokes–Einstein relationship. The effective diameter of the aggregated polymer solution decreased as the DS increased, owing to strong hydrophobic interactions. This result demonstrates that graft copolymer self-aggregation could be affected by the DS of the C_18_ and MA groups, meaning that a higher DS would lead to smaller aggregates and a lower aggregation number because of stronger hydrophobic interactions and tightly packed hydrophobic domains in the self-aggregates. This is particularly evident when comparing DS_M_ 10 and 20 mol%, as in C_18_-M-PHEA and M-PHEA, respectively, in which the effective diameter decreased with the increased DS.

Also, increasing the DS_M_ decreased the critical aggregation concentration (CAC) of grafted copolymers, marked by a significant increase in the ratio (*I*_3_*/I*_1_) of the pyrene emission intensity at a wavelength of 385 nm (*I*_3_) to that at 373 nm (*I*_1_) (Figure 4). According to the result of Figure 4b, the incorporation of C_18_ into the grafted copolymers facilitated the formation of self-assemblies at a low concentration. In summary, the observed variations in effective diameters and critical aggregation concentration (CAC) highlight the pivotal role of the degree of substitution (DS) in governing the self-assembly behavior of C_18_-M-PHEA and M-PHEA. A higher DS, particularly in the C_18_ and MA groups, not only leads to the formation of smaller and more tightly packed aggregates but also lowers the CAC, emphasizing the influence of hydrophobic interactions on the overall self-assembly process. This nuanced understanding elucidates the intricate relationship between DS and the resultant size and structural characteristics, providing valuable insights into the controlled manipulation of self-assembly parameters in graft copolymers.

### 3.2. Spike-like Self-Assembly of C_18_-M-PHEA Integrated with TiO_2_ NPs

Next, in situ incorporation of synthesized TiO_2_ NPs within the graft copolymer self-aggregates was performed using the emulsification method. The morphology of the resulting self-assemblies integrated with TiO_2_ NPs were examined using TEM. As shown in Figure 5a, the synthesized TiO_2_ NPs were aggregated with rod-like structures (rutile phase, Figure 6) in an organic solvent. Figure 5b shows the spherical self-assembled structure formed by C_18_-M-PHEA in an aqueous solution. Similarly, the pure polymers in an aqueous solution, including M-PHEAs and C_18_-M-PHEAs, form spherical self-assemblies. Remarkably, the in situ incorporation of synthesized TiO_2_ NPs within the self-assemblies of graft copolymers, M-PHEA (Figure 5c), and C_18_-M-PHEA (Figure 5d,e) generated TiO_2_ NPs assembled with graft copolymers. Specifically, as illustrated in Figure 5c, the self-assemblies of M-PHEA with synthesized TiO_2_ NPs exhibited a loosely wrapped structure around TiO_2_ NPs. However, as shown in Figure 5e, C_18_-M-PHEA generated well-organized self-assemblies with spike-like TiO_2_ NPs confined by alkyl hydrophobic domains while covered by hydrophilic PHEA, forming a chestnut burr structure in an aqueous solution. The successful in situ incorporation of TiO_2_ NPs within the self-assemblies of graft copolymers, as demonstrated in Figure 5, underscores the versatile nature of the emulsification method in generating well-organized structures. The resulting chestnut burr morphology, particularly that observed in C_18_-M-PHEA, showcases a unique interplay between alkyl hydrophobic domains and hydrophilic PHEA, offering a promising avenue for designing tailored self-assemblies with enhanced functionalities in aqueous solutions.

Furthermore, the cellular toxicity of the graft copolymers, M-PHEA and C_18_-M-PHEA, was found to be significantly lower than that of menthyl anthranilate molecules, as shown in Figure 7. As shown Figure 7b, HDF cells supplemented with MA showed minimal changes in metabolic activities from day 1 to day 3, indicating a stable trend. This observation aligns with the confirmation of slight proliferation in cells at all concentrations. Therefore, to assess the impact of C_18_-M-PHEA supplementation, comparative experiments were conducted up to day 3, providing insights into the contrasting effects compared to MA supplementation. The results show that the metabolic activities of HDFs supplemented with C_18_-M-PHEA increased over time at all concentrations (Figure 7c). It is suggested that the graft copolymers have much lower cellular toxicity than short MA molecules, due to biocompatible poly(amino acid)’s derivative backbone and relatively long chains. Moreover, the notable reduction in cellular toxicity observed with the graft copolymers, M-PHEA and C_18_-M-PHEA, compared to menthyl anthranilate molecules (MA), highlights the biocompatible nature of poly(amino acid)’s derivative backbone. The sustained metabolic activities of HDF cells, even at higher concentrations, suggest the potential of these graft copolymers for biomedical applications. The extended chain length in C_18_-M-PHEA likely contributes to the observed lower cellular toxicity, emphasizing the advantageous role of relatively long chains in enhancing biocompatibility [20]. This bodes well for the prospect of utilizing these graft copolymers in various biomedical and pharmaceutical contexts.

In the culmination of our study, the comprehensive evaluation of UV-blocking properties underscores the significance of the spike-like self-assemblies integrated with TiO_2_ NPs. Through simultaneous UV absorption and scattering analyses, we observed a concentration-dependent effect on UV transmittance, with higher concentrations of C_18_-M-PHEA leading to enhanced UV-blocking efficiency (Figure 8a). Notably, at lower concentrations, UV light penetration was evident, emphasizing the tunable nature of these self-assemblies for desired UV protection levels. The distinct impact of menthyl anthranilate on UVA absorption capacity further accentuates the versatility of our approach in tailoring the UV protection spectrum. As concentration increased, the improved UVA absorption capacity became apparent, reflecting the pivotal role of menthyl anthranilate in augmenting the UVA protection capability of the self-assemblies. Moreover, the UV transmittance of the TiO_2_-loaded C_18_-M-PHEA solution exhibited a remarkable decrease at higher concentrations (Figure 8b). This drastic reduction indicates the effectiveness of our strategy in UV blocking, surpassing the capabilities of unencapsulated C_18_-M-PHEA. The encapsulation of TiO_2_ within C_18_-M-PHEA resulted in a noteworthy nine-fold reduction in UV transmittance compared to its unencapsulated counterpart, underscoring the synergistic UV-blocking effects achieved. This synergy between menthyl anthranilate in the UVA region and TiO_2_ in the UVB region holds significant promise for applications demanding comprehensive UV protection. The chestnut burr structure, formed through the self-assembly of C_18_-M-PHEA integrated with TiO_2_ NPs, further enhances the UV-blocking efficiency, highlighting the potential of our approach for advanced sun-blocking formulations (Appendix A). In conclusion, our findings provide compelling evidence for the efficacy of spike-like self-assemblies with TiO_2_ NPs, presenting a novel strategy for advanced UV protection. The interplay of menthyl anthranilate and TiO_2_, coupled with tunable concentration-dependent behavior, positions these self-assemblies as promising candidates for next-generation sunscreens and UV-blocking materials.

## 4. Conclusions

In conclusion, our study introduces a novel strategy for creating spike-like self-assemblies to enhance UV blocking. The synthesized Poly(2-hydroxyethyl aspartamide) (PHEA) derivatives, including octadecyl chains and menthyl anthranilate (C_18_-M-PHEA), were precisely tailored to control morphology and UV absorption. The in situ integration of polymerized rod-like TiO_2_ within the C_18_-M-PHEA self-aggregates resulted in the formation of spike-like self-assemblies, denoted as TiO_2_@C_18_-M-PHEA, exhibiting a distinctive chestnut burr structure in aqueous solution. Remarkably, our findings reveal that these spike-like self-assemblies, incorporating TiO_2_ NPs, showcased a nine-fold increase in UV protection through simultaneous UV absorption and scattering. This noteworthy enhancement surpasses the UV protection offered by pure TiO_2_ NPs formed using conventional bulk mixing methods. Our work not only provides a novel method for UV protection but also underscores the efficacy of self-assembling poly(amino acid) derivatives integrated with functional nanoparticles for tuning morphology and organization, thereby offering valuable insights for advanced applications in UV-blocking materials.

## Figures and Tables

**Figure 1 polymers-16-00234-f001:**
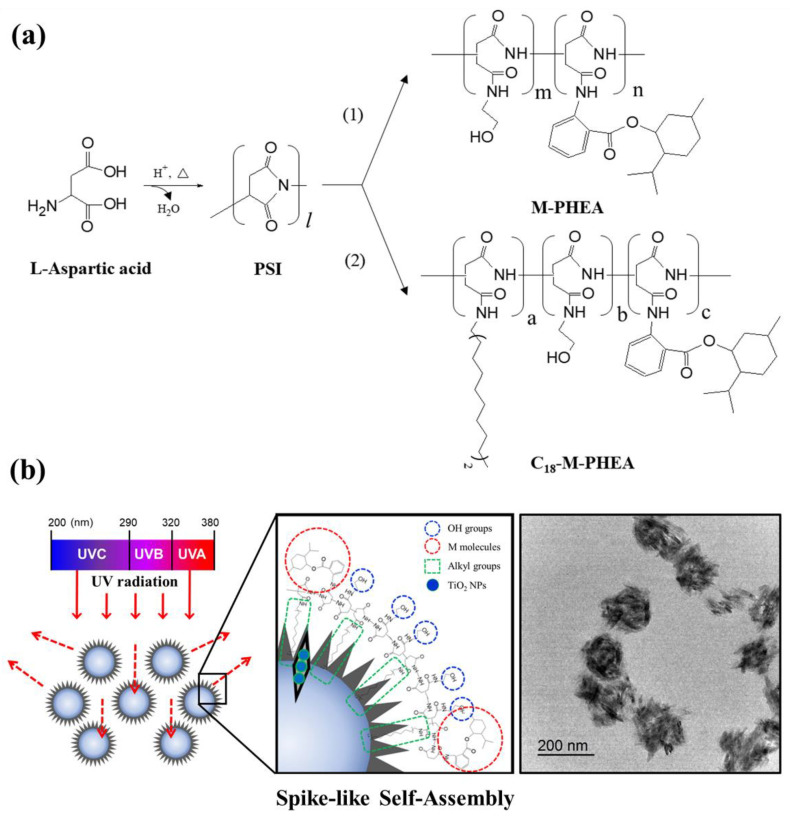
(**a**) Overall synthesis scheme of (1) M-PHEA and (2) C_18_-M-PHEA. (**b**) Schematic representation depicting the formation of spike-like self-assemblies of polyaspartamide integrated with TiO_2_ nanoparticles designed to enhance UV blocking.

**Figure 2 polymers-16-00234-f002:**
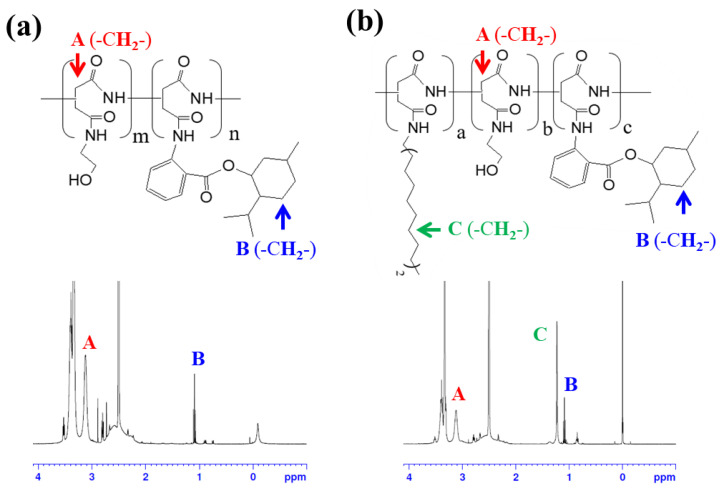
Molecular structure and ^1^H NMR spectra of (**a**) M-PHEA and (**b**) C_18_-M-PHEA, revealing the characteristic features of the synthesized polymers.

**Figure 3 polymers-16-00234-f003:**
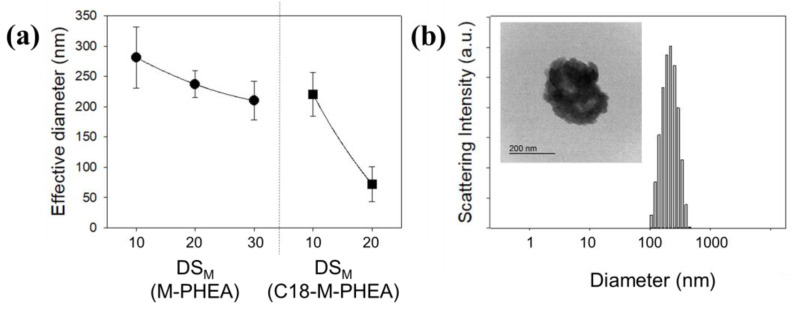
(**a**) Reduction in the effective diameter of M-PHEA and C_18_-M-PHEA with increasing DS_M_ (MA), highlighting the impact of grafting on aggregate size. (**b**) TEM image and size distribution of micelle-like self-aggregates formed from self-assembly of M-PHEA with DS_M_ of 20 mol%.

**Figure 4 polymers-16-00234-f004:**
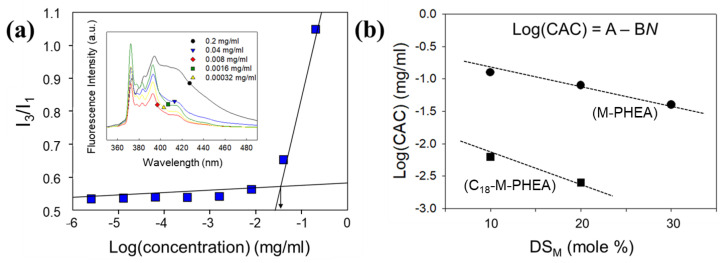
(**a**) Evaluation of the ratio of third-to-first vibrational fine structure (*I*_3_*/I*_1_) and emission spectra (λ_ex_ = 330 nm) of C_18_-M-PHEA. (**b**) Determination of the critical aggregation concentration (CAC) for M-PHEA and C_18_-M-PHEA with varying of DS_M_ (MA).

**Figure 5 polymers-16-00234-f005:**
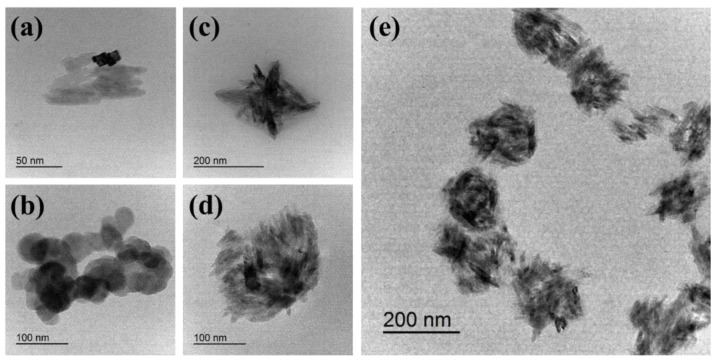
TEM images of (**a**) the synthesized rod-like TiO_2_ nanoparticles, (**b**) the self-assemblies of C_18_-M-PHEA (DS_M_ 20 mol%), (**c**) the self-assemblies of M-PHEA with synthesized TiO_2_ NPs, (**d**,**e**) the spike-like self-assemblies of C_18_-M-PHEA incorporated with TiO_2_ NPs.

**Figure 6 polymers-16-00234-f006:**
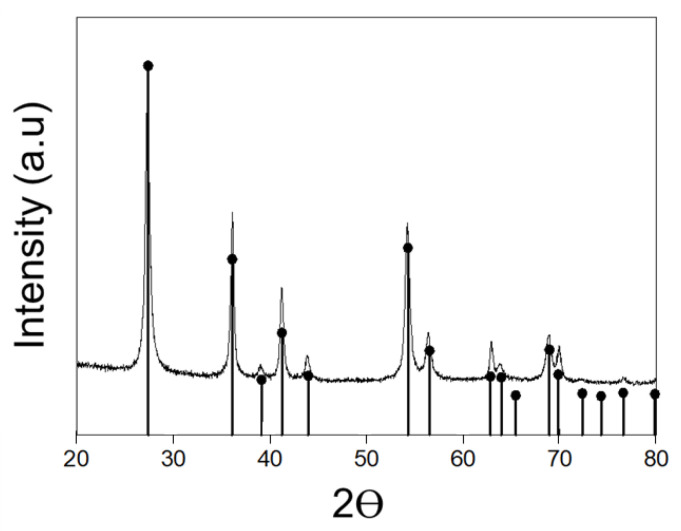
TiO_2_ XRD pattern of self-assemblies incorporated in C_18_-M-PHEA, highlighting the rutile phase.

**Figure 7 polymers-16-00234-f007:**
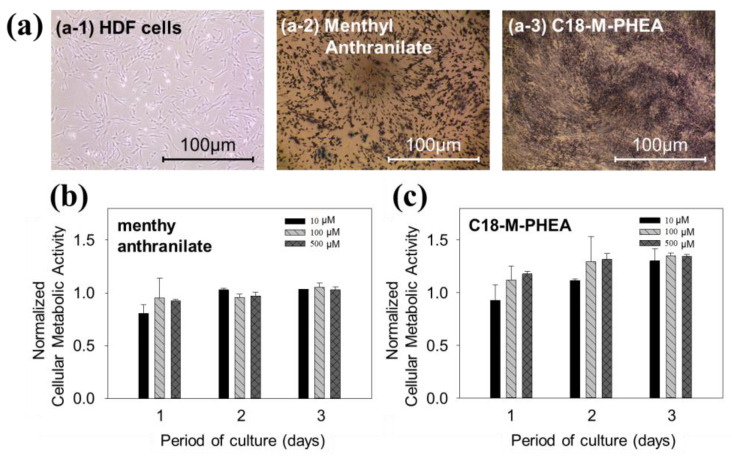
(**a**) Phase-contrast images of HDFs positively stained by MTT assay. The images depict HDFs under the following conditions: (**a-1**) untreated culture, (**a-2**) culture with MA treatment, and (**a-3**) culture with C_18_-M-PHEA treatment. (**b**) Changes in the number of HDFs remained metabolically active with (**b**) MA and (**c**) C_18_-M-PHEA over 3 days. The scale bar represents 100 μm.

**Figure 8 polymers-16-00234-f008:**
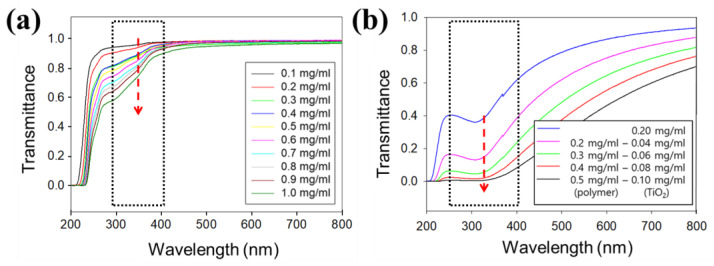
UV-blocking performance. (**a**) UV transmittance data of C_18_-M-PHEA (DS_M_, 20 mol%) versus concentration in the range 0.1–1.0 mg/mL. (**b**) UV transmittance data for TiO_2_ encapsulated in C_18_-M-PHEA (DS_M_, 20 mol%) at a concentration up to 0.5 mg/mL.

**Table 1 polymers-16-00234-t001:** Molecular characterization of C_18_-M-PHEA and M-PHEA series.

Sample	Feed ^a^	DS_C18_ ^b^	DS_M_ ^c^	Number_C18_ ^d^	Number_M_ ^e^	Diameter ^f^
PHEA	100/0	-	-	-	-	Highly soluble
M10-PHEA	90/10	-	4	-	24	281
M20-PHEA	80/20	-	8	-	47	237
M30-PHEA	70/30	-	15	-	88	210
C_18_-M10-PHEA	90/10	4	8	23	47	220
C_18_-M20-PHEA	80/20	5	13	29	76	72

^a^ Feed mole ratio (Succinimide unit/menthyl anthranilate). ^b^ Degree of substitution (mol%) of octadecyl amine determined based on ^1^H-NMR of graft copolymers. ^c^ Degree of substitution (mol%) of menthyl anthranilates determined based on ^1^H-NMR of graft copolymers. ^d^ Number of octadecyl amines per one polymer chain. ^e^ Number of menthyl anthranilates per one polymer chain. ^f^ Effective diameter (nm) obtained by DLS.

## Data Availability

Data are contained within the article.

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
