# Peer review of "A Spike-like Self-Assembly of Polyaspartamide Integrated with Functionalized Nanoparticles"

_polymers, 2024, doi:10.3390/polym16020234_

Round 1

Reviewer 1 Report

Comments and Suggestions for Authors

Dear Authors, 

Your manuscript presents the synthesis of Poly(2-hydroxyethyl aspartamide) (PHEA) substituted with octadecyl chains and menthyl anthranilate (C18-M-PHEA). Then you incorporated rod like TiO2 particles and find out that the shape becomes spike like.  presents more physical explanation about their finding … there is a good presentation of the synthesis, the change of the grafting density, the incorporation of TiO2 but a lack of interpretation of the change in size , shape. Hence, I think if you include more interpretations of what you measured as changes in sizes and chapes it will improve your manuscript and of course cross-check the manuscript for typos and mistakes before resubmission.

1) Please specify the abbreviation DLS (Dynamic light scattering) and include in the text for non-specialists, few sentences explaining how you deduced from it the size.  

2) Please include more physical explanation for the measured change in size and shapes when you change your synthesis ingredients and conditions (degree of substitution, length of the graphted size, quality of the solvent when changing temperature, incorporation of Nanoparticles …)

3) In conclusion, the synthesis and finding are both well-presented but we notes a lack of explanation on why things are so.

4) Please cross-check all the manuscript for typos and mistakes.

Best Regards

Comments on the Quality of English Language

a minor revision with a cross-checking of the manuscript is needed.

Author Response

Dear Editor

the revised manuscript titled “A Spike-like Self-assembly of Polyaspartamide integrated with Functionalized Nanoparticles” to Polymers. We have incorporated the reviewers’ suggestions, which we believe have significantly strengthened the paper. We are grateful for the reviewers’ efforts to help us improve the manuscript. Please find a list of specific changes on the following (attached file) pages. 

Thank you for considering this paper. We look forward to your response. 

Best regards, 

Jae Hyun Jeong, Ph.D.

Associate Professor

Department of Chemical Engineering

Soongsil University

Seoul, 156-743 South Korea

(Tel) +82-2-828-7043

(E-mail) [email protected]

Reviewer 2 Report

Comments and Suggestions for Authors

The literature review (introduction) was treated very superficial. Further it was noted couples of block citation which should be limited to max 3 references together;

Figure 1 was presented at the end of introduction but I found no reference in text about it. Otherwise it should be moved to experiments

“The signals at 2.7, 3.2, and 5.3..” not clear to what refer this. Do you refer to some image ? if so please provide details

“..were analyzed for 24..” this is a weak expression as was not analysed for some amount of time, rather I suspect the samples was analysed after certain period of..

Not clear to what refer for certain figure as “Fig. 1a-2)” cannot be identified anywhere in this manuscript; the same apply for “Fig. 1a-1” and many other;

Some part of results simple repeat or present methodology- for example line 139-142..and many other line in results

Sample preparation for TEM analysis and post processing was not presented in accurate way.

Not clear why you selected period of culture if maximum 3 days

Conclusion are very superficial treated. They should be endorsed with some numerical value from your results.

Comments on the Quality of English Language

needs english improvments 

Author Response

Dear Editor

We are submitting the revised manuscript titled “A Spike-like Self-assembly of Polyaspartamide integrated with Functionalized Nanoparticles” to Polymers. We have incorporated the reviewers’ suggestions, which we believe have significantly strengthened the paper. We are grateful for the reviewers’ efforts to help us improve the manuscript. Please find a list of specific changes on the following (attached file) pages.

Thank you for considering this paper. We look forward to your response.

Best regards,

Jae Hyun Jeong, Ph.D.

Associate Professor

Department of Chemical Engineering

Soongsil University

Seoul, 156-743 South Korea

(Tel) +82-2-828-7043

(E-mail) [email protected]

Reviewer 3 Report

Comments and Suggestions for Authors

The manuscript polymers-2754031 "A Spike-like Self-assembly of Polyaspartamide integrated with Functionalized Nanoparticles" by Han et al. is describes the synthesis of series of polyaspartamide polymer derivatives, TiO2 nanoparticles based on the obtained polymers and the study of toxicity and physical properties. The synthesis was only confirmed by 1H NMR spectroscopy. I think this paper will be of interest to the readers of Polymers. However, the manuscript definitely needs to be corrected. A lot of spectral data is missing.

Questions and comments:

1) The chemical formula of mentyl anthranilate is incorrect.

2) The authors completely misinterpret the NMR spectra of the obtained polymers. The signals of protons B (-CH2-) cannot appear at 1.1 ppm. Please present the full 1H and 13C NMR spectra of the obtained polymers with integral intensities in supplementary materials.

3) Please add the information (abbreviations) of all structures of the obtained compounds with different ratios of reagents (for synthesis) to the experimental part. Absence of clear logic of synthesis and proof of obtained structures is confusing. It is often quite unclear (especially from the figures and their captions) exactly which variants of graft polymers were used. Please correct it so as not to confuse the reader.

4) How correct is it to use the concentration in mol/L for the polymer? How does the concentration of the polymer correspond to the concentration of mentyl anthranilate?

5) Why were not all polymers chosen for biological and physical experiments?

6) The authors almost did not analyze the obtained results. There are no structure-property relationships. The authors did not prove the structure of the polymer-nanoparticle materials from Figure 1b.

7) How does the aggregation properties of the obtained polymers affect their UV-blocking?

8) I recommend comparing the results obtained by the authors with previous results obtained by other scientific groups.

9) Minor changes:

- The manuscript should be re-checked. There are some mistakes and verbs missing.

- Figure 3b. The abscissa axis should be the diameter, not the logarithm of the diameter.

Comments on the Quality of English Language

The manuscript should be re-checked. There are some mistakes and verbs missing.

Author Response

(The authors gave the same response as above.)

Round 2

Reviewer 2 Report

Comments and Suggestions for Authors

.

Comments on the Quality of English Language

.

Reviewer 3 Report

Comments and Suggestions for Authors

I thank the authors for answering my questions and improving the manuscript.

Unfortunately, I couldn't find any supplementary materials. Please add them and the corresponding text from the reviewer's responses to the manuscript.